# Construction of CNT-MgO-Ag-BaO Nanocomposite with Enhanced Field Emission and Hydrogen Sensing Performances

**DOI:** 10.3390/nano13050885

**Published:** 2023-02-27

**Authors:** Xingzhen Liu, Weijin Qian, Yawei Chen, Mingliang Dong, Taxue Yu, Weijun Huang, Changkun Dong

**Affiliations:** Wenzhou Key Lab of Micro-Nano Optoelectronic Devices, Wenzhou University, Wenzhou 325035, China

**Keywords:** CNT-MgO-Ag-BaO nanocomposite, electrophoretic deposition, field emission, tensile test, stability, hydrogen sensing

## Abstract

CNTs and CNT-MgO, CNT-MgO-Ag, and CNT-MgO-Ag-BaO nanocomposites were grown on alloy substrates using an electrophoretic deposition method and their field emission (FE) and hydrogen sensing performances were investigated. The obtained samples were characterized by SEM, TEM, XRD, Raman, and XPS characterizations. The CNT-MgO-Ag-BaO nanocomposites showed the best FE performance with turn-on and threshold fields of 3.32 and 5.92 V.μm^−1^, respectively. The enhanced FE performances are mainly attributed to the reductions of the work function, and the enhancement of the thermal conductivity and emission sites. The current fluctuation of CNT-MgO-Ag-BaO nanocomposites was only 2.4% after a 12 h test at the pressure of 6.0 × 10^−6^ Pa. In addition, for the hydrogen sensing performances, the CNT-MgO-Ag-BaO sample showed the best increase in amplitude of the emission current among all the samples, with the mean I_N_ increases of 67%, 120%, and 164% for 1, 3, and 5 min emissions, respectively, under the initial emission currents of about 1.0 μA.

## 1. Introduction

Carbon nanotubes (CNTs) are very promising as field emitters due to their high aspect ratio, superior mechanical strength, and good conductance as well as their high chemical stability [1,2,3]. At present, CNT-based field emission (FE) technology has been widely applied in electron sources, microwave tubes, traveling wave tubes, X-ray tubes, and electric aerospace propulsion [4,5,6,7,8]. Many methods for the synthesis of CNTs have been investigated, i.e., laser ablation [9], arc discharge [10], chemical vapor deposition (CVD) [11], template methods [12,13], and electrophoretic deposition (EPD) [14].

EPD technology is attractive for the fabrication of CNT-based FE cathodes due to its simple process and low cost [15,16,17]. In the past decades, there have been many reports about CNT FE cathodes using EPD methods [18,19,20]. However, electrophoretic CNT films usually present weak adhesions between CNTs and substrates, leading to obvious deteriorations in FE stability [18]. Metal cations, i.e., Mg^2+^, are usually added to the CNT suspension to enhance the adhesion between CNTs and substrates, which could improve the FE stability [21]. Furthermore, Mg^2+^ together with CNTs can easily transfer to the cathode to form homogeneous CNT films. However, the Mg^2+^ can be changed to MgO after the annealing process, which may weaken the conductivity of pristine CNTs [17,21]. Silver has high electrical and thermal conductivity, and Ag-CNT composites show higher conductivity and thermal conductivity than pristine CNTs [22,23]. Addition of Ag to the mixture solution of CNT and Mg^2+^ might enhance the conductivity and thermal conductivity of CNT-based cathodes. BaO is a thermionic material with a low work function of 1.44 eV [24]. Hence, mixing a small amount of Ba^2+^ with the CNT suspension shows promise to enhance the FE performances of CNT emitters. Therefore, the combination of CNTs with Mg^2+^, silver, and Ba^2+^ is expected to improve FE performances of CNT emitters. Based on the field emission enhancement effect of CNT emitters with gas adsorptions, we developed a new low-pressure gas sensing technique [25,26]. The mini-type field emission sensors are promising for in situ pressure detections in low power consumption and small space applications. The hydrogen sensing effect is mainly related to the amount of gas adsorptions on the emitter surface. However, the gas adsorbents could be de-gassed due to the emitter’s temperature increases due to the high current emission Joule heating effect. MgO can enhance the adhesions between CNTs and substrates, which might benefit CNT stability during the hydrogen sensing performance test. Ag possesses high thermal conductivity, which might reduce the Joule heating to enhance the gas sensing effect. In addition, Barium oxide is a kind of thermionic emission material, which could promote electron emissions with the increase in temperature, benefiting the gas sensing. Therefore, construction of CNT-MgO-Ag-BaO is expected to improve the gas sensing of CNTs.

In this work, CNTs and CNT-MgO, CNT-MgO-Ag, and CNT-MgO-Ag-BaO FE materials were synthesized on alloy substrates using the EPD method and their FE and low-pressure hydrogen sensing performances were investigated. The CNT-MgO-Ag-BaO nanocomposite showed the best FE performance with turn-on and threshold fields of 3.32 and 5.92 V.μm^−1^, respectively. In addition, in comparison with other CNT-based samples, the CNT-MgO-Ag-BaO sample exhibited the best hydrogen sensing performance with the sensing current I_N_ increase of 168% for the 5 min test.

## 2. Materials and Methods

### 2.1. Preparation of CNT-Based Nanocomposite Cathodes on Alloy Substrates

The CNT-based samples were synthesized on alloy sheets (Alfa Aesar, C-276, 0.2 mm thick, Ni:Cr:Fe:Co = 57.5:15.5:6:1.5) using the EPD method [15,16,17]. For the fabrication of CNT electrodes, the raw CNTs (0.10 g L^−1^, Suzhou Tanfeng Nanotech. Port Co., Ltd., Suzhou, China) were first treated by carboxylic processing with concentrated nitric acid, and then the carboxylic CNTs were ultrasonicated in ethanol solution for 5 h, followed by EPD processing to produce the CNT electrodes. For EPD synthesis of CNT-MgO cathodes, MgCl_2_ (0.05 g L^−1^, 99.5%, Aladdin) was added to the ethanol solution with the CNTs. When synthesizing CNT-MgO-Ag cathodes, Ag powder (0.005 g L^−1^, 99.5%, Aladdin) was mixed with carboxylic CNT and MgCl_2_ in ethanol. To produce the CNT-MgO-Ag-BaO cathode, BaCl_2_ crystals (0.026 g L^−1^, 99%, Aladdin) was added to the solution mixture of carboxylic CNT, MgCl_2_, and Ag. During the EPD, the alloy sheet and the stainless steel foil were used as negative and positive electrodes, respectively. The voltage, the processing time, and the electrode distance were 180 V, 8 min, and 1 cm, respectively. Finally, all the samples were annealed in a furnace at 350 °C for 40 min.

### 2.2. Research Methodology

The morphologic and microstructural characteristics of the CNT, CNT-MgO, CNT-MgO-Ag, and CNT-MgO-Ag-BaO cathodes were analyzed by scanning electron microscopy (SEM; JSM-7100F, JEOL, Tokyo, Japan) and high-resolution transmission electron microscopy (HRTEM; JEM-2100, JEOL, Tokyo, Japan). The species of all samples were characterized by X-ray diffraction (XRD; D8 advance, Bruker, Billerica, Germany) and Raman spectroscopy (DXR3, Thermo Fisher Scientific, Waltham, MA, USA). The compositions of the products were analyzed by energy-dispersive X-ray analysis (EDS, Oxford Ultim max, Oxford, UK) and X-ray photoelectron spectroscopy (XPS; Thermo Fisher Scientific escalab250x, Waltham, MA, USA). XPS investigations were conducted with an Al-Kα monochromated X-ray beam under the emission angle of 57° and the spot diameter of 500 μm (chamber pressure: 7 × 10^−8^ Pa). The peak for C 1s at 284.8 eV was used for calibration.

### 2.3. UPS Measurements

The valence band (VB) spectra were measured with a monochromatic He I light source (21.20 eV) and a VG Scienta R4000 analyzer (Thermo Fisher Scientific escalab250x, Waltham, MA, USA). A sample bias of −5 V was applied to observe the secondary electron cutoff (SEC). The work function (ϕ) can be determined by the following formula: ϕ = 21.20 − (BE_SEC_ − E_F_), where BE_SEC_ is the binding energy of the secondary electron cutoff, and E_F_ is the Fermi level.

### 2.4. Tensile Test

The tensile tests for all CNT-based samples were investigated using an Instron 3343 instrument to obtain the adhesion between the films and the alloy substrates, and the experiment detail can be found in our previous report [21]. Briefly, at first, the sample was fixed by a clamp, and then the CNT sample surface was wrapped by tape. During the test, the tape-grabbing film was pulled away until the film peeled off from the alloy substrate. The related data was obtained by the computer control program.

### 2.5. Field Emission and Hydrogen Sensing Test

The field emission and low-pressure hydrogen sensing performances of four types of samples were investigated in a high-vacuum turbo system with a base vacuum of 10^−8^ Pa. The field emission areas were 0.16 cm^2^ with cathode and stainless steel anode distances of 300 μm. The FE current and voltage were controlled by a Keithley 2440 multimeter and Keithley 248 high-voltage supply. The hydrogen sensing test were conducted for the four samples as in our previous reports [25,26,27]. Briefly, a high FE current of ~400 μA generated by applying a voltage was applied first for several minutes to de-gas the adsorbed gas from the surfaces of the samples. Then, the high-purity hydrogen (99.999%) was introduced into the vacuum chamber to maintain a test pressure in the 10^−7^ to 10^−3^ Pa range. Then, a low FE current, typically at the micron ampere level, was applied, and the current variations, which would increase for the sensing emitters, were recorded to acquire the responses to certain hydrogen partial pressures. To obtain reliable pressure sensing data, the normalized current I_N_, i.e., the average value of currents obtained at the end of every 10 s during a certain emission period, was adopted to evaluate the CNT-based samples’ hydrogen sensing performances.

## 3. Results

### 3.1. Morphologic and Structural Characterizations of CNT and CNT-Based Nanocomposites

SEM and XRD characterizations of the CNT-based samples are shown in Figure 1. As shown in Figure 1a, the pristine CNT sample had diameters of 10–30 nm with tangled morphology, probably due to the carboxylic processing. Compared with the pristine CNT sample, the morphologies of CNT-MgO, CNT-MgO-Ag, and CNT-MgO-Ag-BaO samples did not show significant differences (Figure 1b–d). The cross-section of the CNT-MgO-Ag-BaO sample exhibited tight binding structures between the CNT-MgO-Ag-BaO nanocomposite and the alloy substrate (Figure 1e). XRD results for all samples (Figure 1f) indicate that the peaks located at 26.4° were attributed to (002) planes of CNTs [21,28], and the other three peaks at 43.5°, 50.5°, and 74.3° belonged to the (111), (200), and (220) planes of the alloy substrates, respectively [29]. For the CNT-MgO-Ag and CNT-MgO-Ag-BaO samples, two additional peaks at 38.2° and 64.5° were observed, which could be attributed to metal Ag [21,22]. For the CNT-MgO-Ag-BaO composite, the characteristic peaks of BaO did not appear [30,31], probably due to the small amount of BaO or the existence of amorphous BaO; similar cases could be observed in the NCNT−Pd composite [15].

Raman and EDS results for the CNT-based samples are shown in Figure 2. As shown in Figure 2a, the Raman spectra of the four samples exhibited the same CNT characteristic peaks, i.e., the D band (1346 cm^−1^), G band (1587 cm^−1^), and G’ (2696 cm^−1^), respectively. The D band is related to the defect peak from the disordered CNT structures, whereas the G band is the graphite degree peak [15,28]. The ratio of the D band peak to the G band peak, i.e., I_D_/I_G_, is often used to define the structural disorder. In comparison with the pristine CNT, the I_D_/I_G_ of each of the other three CNT-based composites was about the same, implying that CNT crystal structures did not change during composite productions. EDS characterizations were investigated for all samples, as shown in Figure 2b. For the pristine CNTs, the EDS exhibited the signals of C, O, Fe, Cr, and Ni, as expected. The elements of Fe, Cr, and Ni belong to the alloy substrate (C-276). The O element was mainly from the CNT surface contamination from the carboxylic processing and exposure to the air. For the CNT-MgO sample, a Mg signal was observed, which was from MgO after the annealing treatment. For the CNT-MgO-Ag sample, a Ag signal could also be seen from the Ag powder. However, for the CNT-MgO-Ag-BaO sample, a Ba signal was detected.

TEM and EDS mapping images of CNT-MgO-Ag-BaO nanocomposites are shown in Figure 3. As can be seen, the diameters of the CNTs were about 10–30 nm. Curved graphite layers could be observed, showing structural defects in CNTs (Figure 3b), in agreement with the XRD and Raman characterizations. An HRTEM image of the CNT-MgO-Ag-BaO sample suggests that the Ag nanoparticles, with the diameters of ~4nm, were attached on the surface of the CNTs with a lattice spacing of about 0.238 nm attributed to the d_111_ plane of Ag (Figure 3c) [32], agreeing with the XRD results. The compositions of the CNT-MgO-Ag-BaO sample were obtained using the EDS mapping characterizations (Figure 3d–i), showing C, O, Mg, Ag, and Ba elements, as expected. As shown in Figure 3d, the outline of the carbon nanotubes was shown by dotted lines. The same dotted lines were also shown in Figure 3e–i; the signals of C, O, Mg, and Ag can be clearly observed on the surface of the carbon nanotubes. It can be seen that the Ba signal is weaker than that of the other elements, probably due to the low concentration or insufficient uniformity. In addition, some BaO components might fall off the surface of the CNTs during the TEM sample preparation due to the effect of ultrasound.

XPS characterizations were further investigated for all samples with the anticipated signals of C, O, Mg, Ag, and Ba, as shown in Figure 4a. For the Mg 1s (Figure 4b), the peaks located at 1304.2 eV belong to the Mg-O bonds, rather than the metallic Mg at ~1303 eV for the CNT-MgO, CNT-MgO-Ag, and CNT-MgO-Ag-BaO samples [15,17]. For the O 1s spectra (Figure 4c), two peaks at 532.98 eV and 532.13 eV were attributed to the C-O and C=O bonds for all samples [21]. In addition, a peak at 530.95 eV could be observed for the CNT-MgO, CNT-MgO-Ag, and CNT-MgO-Ag-BaO samples, which was attributed to the Mg-O bond [21]. For the CNT-MgO-Ag-BaO sample, a Ba-O bond could be seen with the BE value of 529.87 eV [30]. For the Ag 3d, the binding energies (BE) located at 374.4 eV and 368.4 eV correspond to the Ag 3d_3/2_ and Ag 3d_5/2_, respectively, which belong to the metallic silver [15,28]. For the Ba 3d, two peaks located at 341.2 and 335.9 eV correspond to Ba 3d_3/2_ and Ba 3d_5/2_, respectively, suggesting the existence of BaO in the CNT-MgO-Ag-BaO composite [30].

### 3.2. Field Emission Performance Test

Field emission performances, including current density-emission field (J-E), Fowler-Nordheim (F-N) curves, and FE stability, as well as the film adhesion strengths from the tensile tests of all samples, were investigated, as shown in Figure 5. Because the two curves for the CNT-MgO-Ag and the CNT-MgO-Ag-BaO samples are very close to each other, repeated tests were necessary and further investigations should be devoted to optimizing the contents of Ag and BaO. The turn-on fields E_to_ and threshold fields E_thr_ are defined as the electric fields required to generate emission current densities by 10 μA.cm^−2^ and 10 mA.cm^−2^, respectively. The FE performances of the CNT-based samples were repeated three times to confirm the repeatability of the FE characteristics and the average values were obtained for comparison (Table 1). As shown in Appendix A, all four samples presented high FE repeatability, and the J-E curves do not deviate from each other evidently after three rounds of FE tests. In comparison with other CNT-based samples, the CNT-MgO-Ag-BaO sample showed the best FE performance with the lowest turn-on field of 3.32 V.μm^−1^ and the lowest threshold field of 5.92 V.μm^−1^ (Figure 5a, see Table 1). The CNT−MgO−Ag cathode displayed better E_to_ and E_thr_ than the CNT and CNT−MgO samples, which could be attributed to the lower surface resistance [21] and enhanced emission sites. Compared with the CNT-MgO-Ag sample (Figure 6a–c, 21.20 – 16.66 = 4.54 eV), the CNT-MgO-Ag-BaO could have more emission sites and a lower work function (Figure 6d–f, 21.20 – 16.95 = 4.25 eV) due to the introduction of BaO, and similar cases can also be observed in the CNT-BaO, leading to better FE properties [33]. As shown in Figure 5b, the Fowler-Nordheim (F-N) curves for all samples exhibited linear relationships, suggesting that all CNT-based emitters obey the ideal FE processes. The field enhancement factor (β) is considered as an evacuation parameter for cathodic material during the FE process, which can be calculated based on the F-N formula: β = −6.83×10^3^×ϕ^3/2^/slope [13,34], where β is the field enhancement factor, and ϕ is the work function of the cathode material; the slope can be obtained from the F-N plot. The work function of the pristine CNT, CNT-MgO, CNT-MgO-Ag, and CNT-MgO-Ag-BaO emitters are regarded as 4.68, 4.65, 4.54, and 4.25 eV (see Figure 6), respectively. According to the F-N formula, the field enhancement factor of the pristine CNT, CNT-MgO, CNT-MgO-Ag, and CNT-MgO-Ag-BaO samples are estimated to be about 1117, 1380, 1812, and 1417, respectively. The emission stability of all CNT-based field emitters was investigated, as shown in Figure 5c. For the pristine CNTs, after a short rise, the current reduced by 12.3% after 12 h emission. In comparison with the pristine CNT emitter, the CNT-MgO sample showed better FE stability with a current drop of 6.6%, probably attributable to better adhesions with the tension stress value of 0.92 N (Figure 5d), which is higher than that of the CNTs (0.66 N, Figure 5d). Compared with the CNT-MgO sample, the CNT-MgO-Ag sample displayed a lower current fluctuation (4.8%), and benefited from the introduction of Ag with good thermal conductivity. For the CNT-MgO-Ag-BaO sample, the current fluctuation (2.4%) was better than that of the CNT-MgO-Ag sample, probably due to the effect of the thermionic emission material of BaO, which could transfer the Joule heat during the FE process more efficiently to reduce the damage of the CNT−based sample.

To investigate variations of the work function for all the CNT-based samples, UPS measurements were conducted, as shown in Figure 6. The sample bias of −5 V was applied to obtain the secondary electron cutoff (SEC), which is shown in Figure 6a,d,g,j, corresponding to the CNTs, and the CNT-MgO, CNT-MgO-Ag, and CNT-MgO-Ag-BaO composite samples, respectively. All the values of the Fermi level (E_F_) were calibrated with the value of 0 eV (Figure 6b,e,h,k), as further displayed in Figure 6c,f,i,l, which were partially enlarged (near E_F_) from Figure 6b,e,h,k, respectively. Therefore, the work function (ϕ) can be determined by the difference between the photon energy (21.20 eV) and the binding energy (BE) of the secondary cutoff edge (SEC), i.e., ϕ = 21.20 − BE_SEC_ [35]_._ Take the CNT sample, for example (Figure 6a–c). Figure 6a shows that the value of BE_SEC_ is 16.52 eV, thus the work function of the CNTs is 4.68 eV (21.20 − 16.52 = 4.68). Similarly, the work functions of the CNT-MgO, CNT-MgO-Ag, and CNT-MgO-Ag-BaO samples can also be calculated with the values of 4.65, 4.54, and 4.25 eV, respectively. Therefore, the construction of the CNT-MgO-Ag-BaO composite can effectively reduce the work function of the CNT sample.

### 3.3. Hydrogen Sensing Performance Test

The hydrogen sensing properties were studied in the pressure range of 10^−^^7^ to 10^−^^4^ Pa for all samples under the initial emission currents of 1.0 μA, as shown in Figure 7. To ensure the sensing reliability under low emission current conditions, the normalized average current I_N_ was employed [26,27]. In addition, to claim the better performance of the CNT-MgO-Ag-BaO with respect to other CNT-based samples, error bars associated with the I_N_ increase are necessary. As shown in Figure 7, the emission currents increased with the increase in hydrogen partial pressure for all samples. When the hydrogen partial pressures were less than 10^−^^5^ Pa, the emission currents increased slightly for the CNT and the CNT-MgO samples (Figure 7a,b), but the emission currents of the CNT-MgO-Ag sample increased more quickly (Figure 7c), probably due to the catalysis effect of Ag [36]. As shown in Figure 7d, the CNT-MgO-Ag-BaO sample showed the highest current-increase amplitude among all the samples, with the I_N_ increase of 67%, 120%, and 164% for 1, 3, and 5 min emissions, respectively. Differences in the shapes of the sensing curves for different samples might be attributed the following two reasons. Firstly, the hydrogen detections were conducted in low-pressure vacuum environments based on the gas adsorptions on field emitters, and the gas adsorption results in the reduction of the work function for the emitter material, leading to the emission current increase, and thus the sensing effect. Therefore, the sensing curves are highly related to the types of cathodic materials, and may present different shapes. Secondly, the gas sensing current I_N_ is a normalized current rather than a direct output current after applying the electric field, and I_N_ can be regarded as the average value of currents obtained at the end of every 10 s during a certain emission period. Thus, both linear and nonlinear curves may appear. The hydrogen sensing effect is mainly related to the amount of gas adsorptions on the emitter surface. However, the gas adsorbents could be de-gassed from the emitter temperature rises due to high current emission Joule heating effect. Ag possesses high thermal conductivity, which might slow down the Joule heating to enhance the gas sensing effect. In addition, Barium oxide is a kind of thermionic emission material, which could promote electron emissions with the temperature rise, improving the gas sensing. Although the turn-on and threshold fields of the unmodified CNTs in our previous reports [26,27] are lower than the ones obtained in this work for the CNT-MgO-Ag-BaO sample, the emission stability of CNT-MgO-Ag-BaO is better than that of the unmodified CNTs. In addition, the emission current of the CNT-MgO-Ag sample showed a higher current-increase amplitude than that of the unmodified CNTs in the lower pressure range (less than 10^−^^5^ Pa). Therefore, the CNT composite FE cathodes provide promising approaches to developing practical FE cathodes with adequate low-pressure hydrogen sensing effects.

## 4. Conclusions

In this paper, CNT, CNT-MgO, CNT-MgO-Ag, and CNT-MgO-Ag-BaO field emitters were grown on alloy substrates using the electrophoretic deposition method, and their field emission (FE) and hydrogen sensing performances were investigated. The turn-on field and threshold field of the CNT-MgO-Ag-BaO emitter are 3.32 and 5.92V.μm^−1^, respectively, which are both lower than those of the other CNT-based emitters. The improvements of FE performances are mainly attributed to the reduction of the work function, and the enhancements of the thermal conductivity and emission sites. The current fluctuation of CNT-MgO-Ag-BaO nanocomposites was only 2.4% after a 12 h test at the pressure of 6.0 × 10^−6^ Pa. Meanwhile, for the hydrogen sensing, the CNT-MgO-Ag-BaO sample displayed the highest current-increase amplitude among all the samples, with I_N_ increases of 67%, 120%, and 164% for 1, 3, and 5 min emissions, respectively. The excellent field emission and hydrogen sensing performances suggest that the facile growth of CNT-MgO-Ag-BaO emitters might provide a promising approach to developing practical FE cathodes.

## Figures and Tables

**Figure 1 nanomaterials-13-00885-f001:**
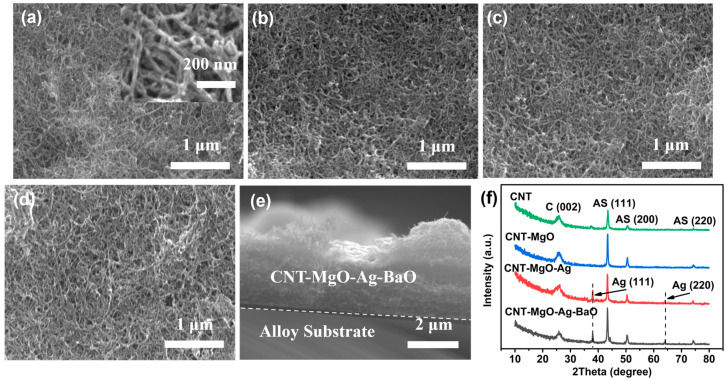
SEM images of the pristine CNTs and CNT-based nanocomposite films. (**a**) The pristine CNTs; (**b**) CNT-MgO composite; (**c**) CNT-MgO-Ag composite; (**d**) CNT-MgO-Ag-BaO composite; (**e**) cross-section of the CNT-MgO-Ag-BaO composite film; (**f**) XRD results of the pristine CNTs and CNT-based nanocomposites.

**Figure 2 nanomaterials-13-00885-f002:**
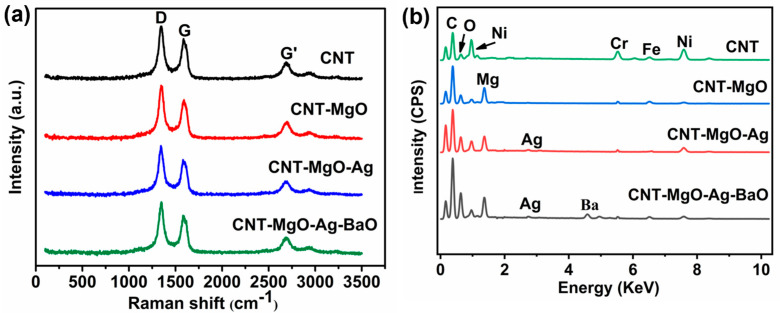
(**a**) Raman and (**b**) EDS results of the pristine CNTs and CNT-based nanocomposites.

**Figure 3 nanomaterials-13-00885-f003:**
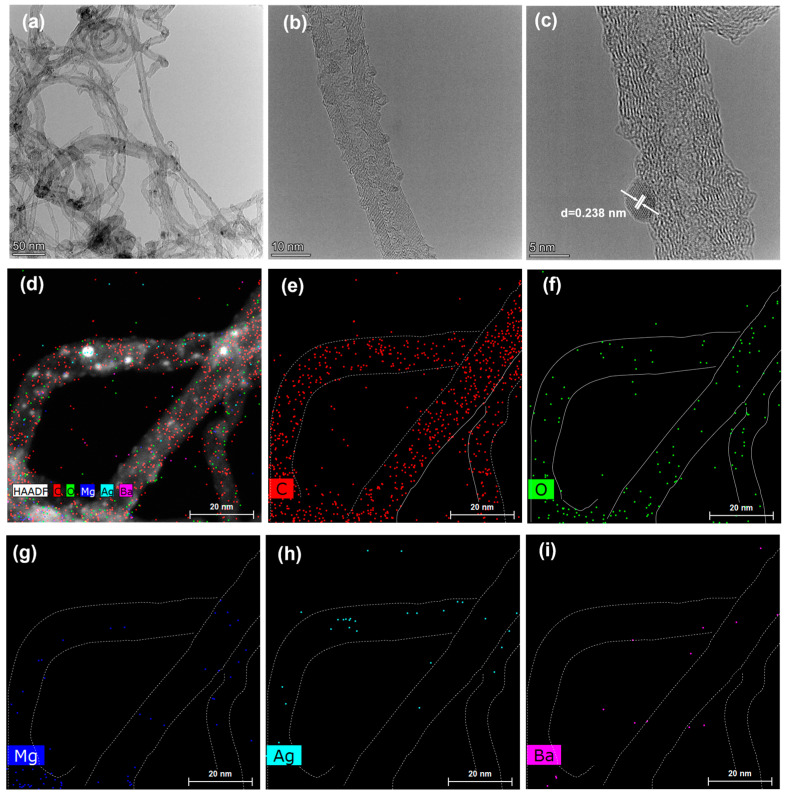
TEM images and EDS analysis of the CNT-MgO-Ag-BaO composite. (**a**) Low-resolution TEM image of the CNT-MgO-Ag-BaO sample, and HRTEM images of (**b**) CNT and (**c**) Ag nanoparticle; (**d**–**i**) EDS mappings of C, O, Mg, Ag, and Ba elements from the CNT-MgO-Ag-BaO composite.

**Figure 4 nanomaterials-13-00885-f004:**
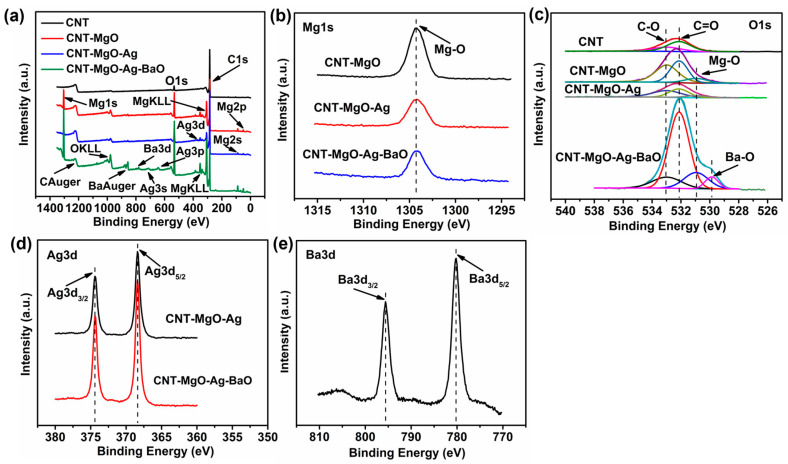
XPS spectra of the CNT, CNT-MgO, CNT-MgO-Ag, and CNT-MgO-Ag-BaO composite samples. (**a**) Survey; (**b**) Mg1s; (**c**) O1s; (**d**) Ag3d; (**e**) Ba3d.

**Figure 5 nanomaterials-13-00885-f005:**
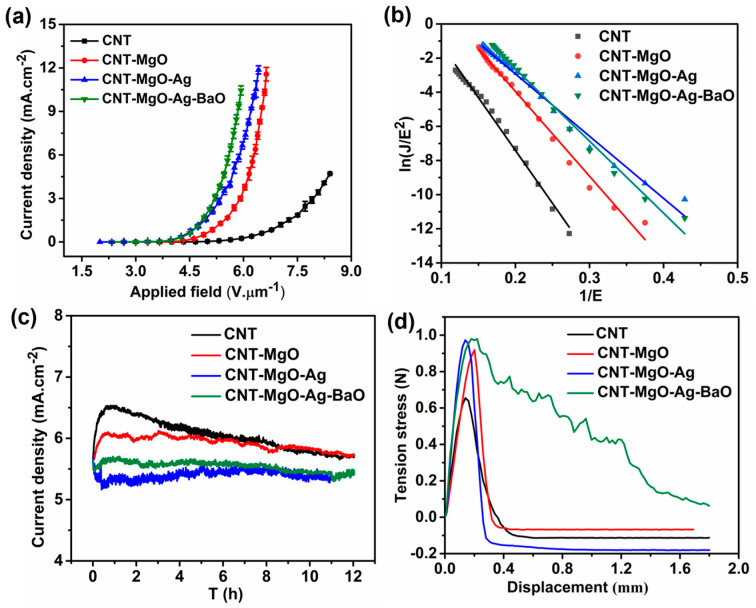
Field emission performances and tensile tests of the CNT and CNT-MgO, CNT-MgO-Ag, and CNT-MgO-Ag-BaO composite samples. (**a**) J-E curves; (**b**) F-N curves; (**c**) FE stability tests; (**d**) Tensile tests.

**Figure 6 nanomaterials-13-00885-f006:**
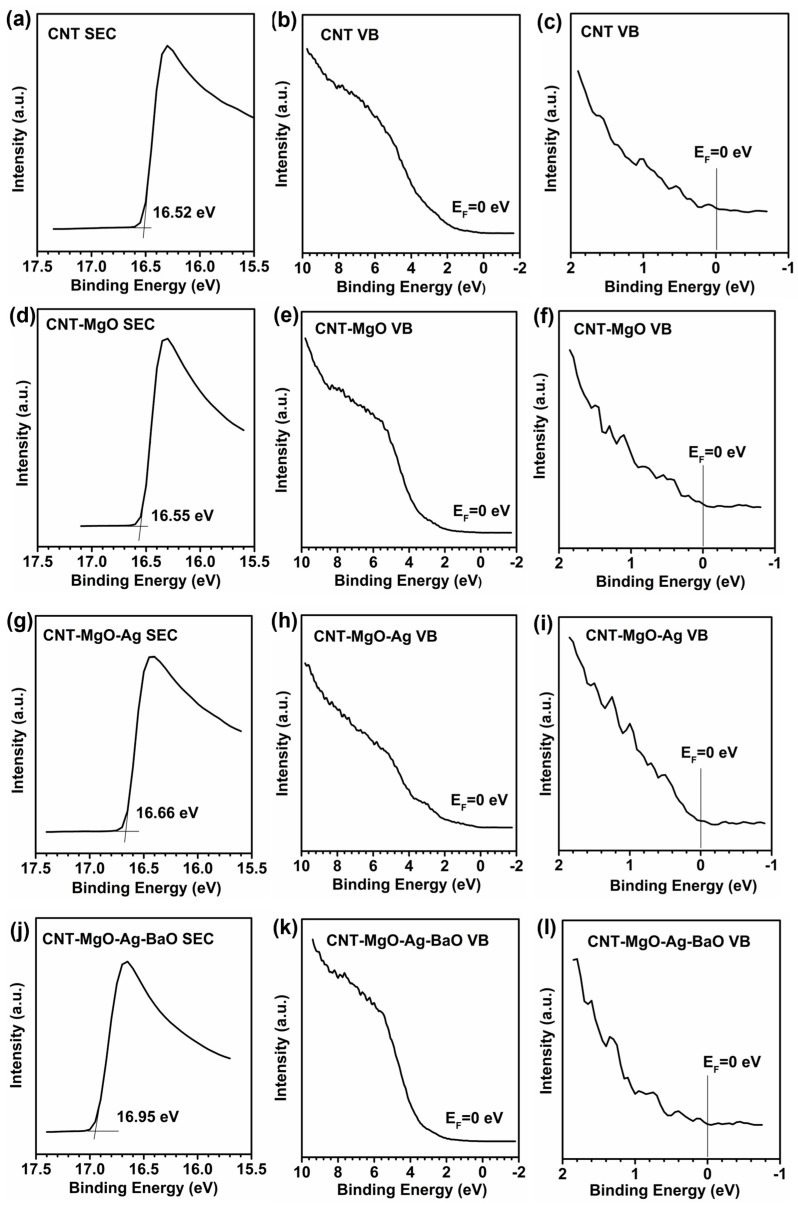
UPS measurements of (**a**–**c**) the CNT, (**d**–**f**) CNT-MgO, (**g**–**i**) CNT-MgO-Ag, and (**j**–**l**) CNT-MgO-Ag-BaO composite samples. Note: SEC is the secondary electron cutoff, VB is the valence band, and E_F_ is Fermi level.

**Figure 7 nanomaterials-13-00885-f007:**
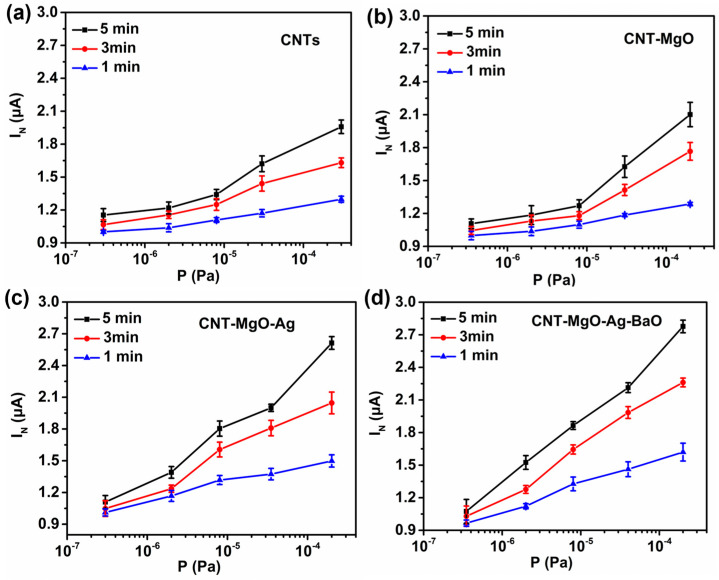
Pressure sensing performances of the CNTs and CNT-based composite samples. (**a**) CNTs; (**b**) CNT-MgO; (**c**) CNT-MgO-Ag; (**d**) CNT-MgO-Ag-BaO.

**Table 1 nanomaterials-13-00885-t001:** Comparison of FE parameters of CNT−based samples.

Sample	Turn-on Field (V/µm)	Threshold Field (V/µm)	β
CNT	4.99	/	1117
CNT−MgO	3.99	6.59	1380
CNT-MgO-Ag	3.65	6.33	1812
CNT-MgO-Ag-BaO	3.32	5.92	1417

## Data Availability

Not applicable.

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
