# Peer review of "Construction of CNT-MgO-Ag-BaO Nanocomposite with Enhanced Field Emission and Hydrogen Sensing Performances"

_nanomaterials, 2023, doi:10.3390/nano13050885_

Round 1

Reviewer 1 Report

In this manuscript, the authors explore CNT nanocomposites incorporating MgO, Ag, and BaO. The field emission and hydrogen sensing performance of the nanocomposites were examined. The work is well organized/written and falls within the scope of Nanomaterials. I would recommend publication – I just have one minor comment.

1.     I think it would be better to use a common x and y scale for all the plots in Figure 7 a) to d) to illustrate the improved performance with incorporation of MgO, Ag, and BaO.

Author Response

Changkun Dong

Professor, Electronic Engineering

Director, Institute of Micro-nano Structures & Optoelectronics

Wenzhou University

Chashan University Town, Wenzhou

Zhejiang 325035, P. R. China

Tel: 086-577-86689067, Email: dck@wzu.edu.cn

Feb. 12, 2022

Re: Response to reviews’ comments

Dear Editor/Reviewers:

Thank you for reviewing our manuscript entitled “Construction of CNT-MgO-Ag-BaO nanocomposite with enhanced field emission and hydrogen sensing performances” (Ref. No.: nanomaterials-2192903)!

We have studied the comments carefully. The comments are valuable for revising our manuscript. We have made corrections following the comments. Revised portions are highlighted in red type in the revised manuscript. The main corrections and the point-to-point responses to reviewers’ comments are discussed as follows.

Once again, we appreciate very much the great comments from the reviewers and the support from the editor! We hope that the responses will meet with the acceptance.

Very sincerely yours,

Changkun Dong

Reviewer 2 Report

The article presents CNTs nanocomposites for field emission and hydrogen sensing. More details on how the characterization was done is needed and the more detail on the interpretation of the results is needed. Some characterization results are poor quality. A comparison with other emitter and hydrogen sensing is needed to correctly evaluate this work results.     Below I have more comments about this issue and suggestion on how to improve the manuscript.

  1. Materials and Methods section
    1. Why the acquisition parameters were only given for XPS and not for the other technic?
    2. Which EDS was used?
    3. Which alloy and composition was used? Why this alloy?
  2. Results section
    1. How the cross section CNT-MgO-Ag-BaO sample was obtained?
    2. Fig 1 e: What is the thin dark layer between the alloy and CNT?
    3. Fig1 f: Indicate where the Ba peaks should appear? Could be that BaO phase is amorphous?
    4. Fig 2b: Why the Ni, Cr and Fe signal is smaller for the CNT nanocomposite than for the pristine CNT?
    5. Fig 3 e-i: EDS images are just noise, you cannot publish them. They should show images related to the change of composition shown in the HAADF image.
    6. Fig 4 a: Why some signal are not identified? Which element/bond?
    7. Why the tensile test is not described in the Methods section?
    8. Fig 5a: does not show the turn on field value for each sample.
    9. Figure 6 is not well introduced in the text (mixed with description of Fig 5)
      1. What an UPS measurement? It is not described in the Methods section.
    10. Fig 7: Are each measurement repeated? What are the error bar on the measurement?
      1. Maybe plot each sample with the same Y axis minimum and maximum for easier comparison between them.
  3. Conclusions
    1. No comparison with other emitter and hydrogen sensing of this work results.

Author Response

(The authors gave the same response as above.)

Reviewer 3 Report

The paper entitled “Construction of CNT-MgO-Ag-BaO nanocomposite with enhanced field emission and hydrogen sensing performances”, by Xingzhen Liu and co-authors, proposes an analysis on the field emission and hydrogen sensing performance of carbon nanotubes (CNTs) and their composites.

The abstract, introduction and conclusions do not clearly expose the point and the aim of the paper. The introduction itself does not contain any information on the reason why the CNTs composites have been chosen for hydron sensing. There is no ligand between the different steps of the CNT functionalization and in-depth analysis on the choices and reasons leading to one compound instead of another.

The authors underline the importance of Barium (Ba) because of its low work function (WF) anyway there is no calculation of the CNTs WF before and after BaO addition.

Following the point-by-point comments on the manuscript and experimental analysis.

1.      Reference 19 at page 1 is reported to support the use of cations Mg2+ for a better adhesion between CNTs and the substrate. On the contrary in reference 19 there is no mention about cations. The authors should replace the reference with a more specific one.

2.      The whole manuscript lacks of explanation on the choice of the compounds used for the CNT blenders, as well as, regarding the hydrogen sensing. Why did the author choose to investigate the H2 sensing with CNTs?

3.      In Figure 1f the authors report the XRD results for the CNTs and their composites. The XRD spectra of CNT, CNT-MgO and CNT-MgO-Ag-BaO show C(002) peak around 26.4°. Nevertheless CNT-MgO-Ag displays only a small shoulder around that angle, which then becomes a peak after the adding of BaO. Can the author argue on the reduction of the C(002) peak in the CNT-MgO-Ag composite?

4.      Figure 2b shows the EDS spectra for the four systems. The authors describe the purple and green curves (CNT-MgO-Ag and CNT-MgO-Ag-BaO, respectively) as “For the CNT-MgO-Ag sample, Ag signal could also be found from the Ag powder. While for the CNT-MgO-Ag-BaO sample, the Ba signal was detected.”

According to the author description an Ag signal can be found in the CNT-MgO-Ag, but not in the CNT-MgO-Ag-BaO. The authors should explain for what reason the Ag signal is not present in the green spectrum, where the Ba component is the only present.

5.      Considering the diameter of the Ag nanoparticles (~4nm) they should be visible from SEM images too. Figure 1c should be taken in a higher magnification to prove the presence of the Ag nanoparticles on the CNTs.

6.      In the Field emission performance test section, the authors report the turn on and the threshold field values for the best performing system (CNT-MgO-Ag-BaO), 3.4 Vµm-1 and 4.7 Vµm-1.

From Figure 5a there is no significant difference between the field emission curve acquired for the CNT-MgO-Ag-BaO and the CNT-MgO-Ag system. The two curves are very close to each other and to compare the performance between the fours system the author should report the values with the relatives errors.

7.      As stated in the introduction “BaO is a thermionic material with low work function of 1.44 eV [24]. Hence, mixing of small amount of Ba+2 with the CNT suspension is promising to enhance the FE performances of CNT emitters”, did the author calculate the change of the CNT WF before and after BaO addition?

8.      Figure 6 (UPS measurements) is never commented and mentioned in the text. Each figure in the manuscript must be deeply described and explained.

9.      The chamber pressure is not a quantification of the hydrogen presence. For sensing application, the detectivity of the device should be referred to ppm (particles per million). Can the authors quantify the hydrogen in ppm and report literature values to which compare the results of their manuscript? In other words, what is the state-of-the-art for hydrogen sensing with CNTs?

10.  Figure 7 displays the hydrogen pressure performances of the four systems, yet the trends of the curves for the systems is very different. CNT and CNT-MgO show a power-like behavior while CNT-MgO-Ag and CNT-MgO-Ag-BaO a more linear trend. Do the author have an explanation for this behavior change? They should comment about that in the main text.

11.  In absolute values there is no significant difference between CNT-MgO-Ag and CNT-MgO-Ag-BaO currents. To claim the better performance of the CNT-MgO-Ag-BaO respect to the CNT-MgO-Ag system error bars associated with the IN increase are required.

12.  Considering references 26 and 27 similarities between the manuscript and the two cited works arise.

In reference 26 (Vacuum 2023, 207, 111663) the authors tested multi-walled CNTs obtaining interesting turn on and threshold fields (1.67 Vµm-1 and 2.9 Vµm-1) for the unmodified CNTs lower than the ones obtained for the CNT-MgO-Ag-BaO. Can the author underline where the improvement is achieved by their new system (CNT-MgO-Ag-BaO).

In reference 27 (Carbon 2017, 124, 669) the authors provide an example of low pressure hydrogen sensing with CNTs. Following the same procedure of the submitted manuscript, the authors shows with the bare CNTs similar results to the ones obtained with the blended system CNT-MgO-Ag-BaO.

In the present paper the current value varies from ~1 µA at 10-7 Pa to ~2.7 µA at 10-4 Pa (a factor 3) while in the previous paper (ref 27) they display a current change from ~0.1 µA at 10-7 Pa to ~0.34 µA at 10-4 Pa, a factor 3 as well.

Therefore, where is the importance in using CNTs composites if the same results can be achieved with CNTs prepared as in ref 29?

Considering the previous comments, concerns, and suggestions, and the lack of novelty of the paper, I strongly believe that this work is not suitable for publication. Therefore, I suggest rejection without any further revision.

Author Response

(The authors gave the same response as above.)

Reviewer 4 Report

See attached.

Author Response

(The authors gave the same response as above.)

Round 2

Reviewer 3 Report

The authors did not deeply answer the referee questions and comments.  

COMMENT 1

The answer to comment 9 is not acceptable. The authors explain the choice of quantifying the H2 presence citing themselves [25-27].

"it is difficult to give a quantification of the hydrogen presence, the partial pressure of hydrogen is often used to investigate the hydrogen sensing [25-27]."

Different citations, from our groups, are required.

The authors answered to comment 9 citing themselves

COMMENT 2

The answer to comment 4 requires additional experiments, the low concentration or the insufficient uniformity of the samples are not a sufficient explanation. The sample should be uniform, also considering that has been synthesized adding BaO to the previous CNT-MgO-Ag where the Ag peak is present.

The authors should test again the sample in different regions and add new experimental data

"Generally, Ag signal should be found in the CNT-MgO-Ag-BaO as that of the CNT-MgO-Ag sample. However, Ag signal was not be observed in the CNT-MgO-Ag-BaO, probably due to low concentration or insufficient uniformity for the testing area."

COMMENT 3

“The variation of the C(002) peak in the CNT-MgO-Ag composite is probably due to the increasing short CNTs during the synthesis process, leading to the increasing disorder of the CNTs.”

What is the main message of this sentence. What other proof do the authors have for the shrinking of the CNTs?

COMMENT 4

The FE performances of the CNT-based samples were repeated for three times to confirm the repeatability of the FE characteristics and the average values were obtained for comparison (Table 1). As shown in Fig. S1 (see supplementary material SI-1), all four samples presented high FE repeatabilities, and the J-E curves do not deviate from each other evidently after three round FE tests

The repeatability of the experiments and the given error bars should be proved and calculated on different systems not only repeating the experiments on the same device for three times.

COMMENT 5

“Firstly, the hydrogen detections were conducted in the low pressure vacuum environments based on the gas adsorption, which is different from the hydrogen detection in the air.”

The first part of this answer is confusing. Why are the authors referring to the detection in air if they have done all the measurements in vacuum? The difference in shape of the curves has been evidenced for the four systems tested from 10-7 to 10-4 Pa therefore the air is not playing any role, and this first part of the answer cannot be considered as an explanation.

In conclusion I do not believe that the paper is ready for publication.

Author Response

Changkun Dong

Professor, Electronic Engineering

Director, Institute of Micro-nano Structures & Optoelectronics

Wenzhou University

Chashan University Town, Wenzhou

Zhejiang 325035, P. R. China

Tel: 086-577-86689067, Email: dck@wzu.edu.cn

Feb. 21, 2022

Dear Editor/Reviewers:

Thank you for reviewing our manuscript entitled “Construction of CNT-MgO-Ag-BaO nanocomposite with enhanced field emission and hydrogen sensing performances” (Ref. No.: nanomaterials-2192903)!

We have studied the comments carefully. The comments are valuable for revising our manuscript. We have made corrections following the comments. Revised portions are highlighted in red type in the revised manuscript. The main corrections and the point-to-point responses to reviewers’ comments are discussed as follows.

Once again, we appreciate very much the great comments from the reviewers and the support from the editor! We hope that the responses will meet with the acceptance.

Very sincerely yours,

Changkun Dong

Reviewer 4 Report

The manuscript is ready for publication. 

Author Response

Changkun Dong

Professor, Electronic Engineering

Director, Institute of Micro-nano Structures & Optoelectronics

Wenzhou University

Chashan University Town, Wenzhou

Zhejiang 325035, P. R. China

Tel: 086-577-86689067, Email: dck@wzu.edu.cn

Feb. 21, 2022

Dear Editor/Reviewers:

Thank you for reviewing our manuscript entitled “Construction of CNT-MgO-Ag-BaO nanocomposite with enhanced field emission and hydrogen sensing performances” (Ref. No.: nanomaterials-2192903)!

Once again, we appreciate very much the great comments from the reviewers and the support from the editor! 

Very sincerely yours,

Changkun Dong